# Characterization of a Novel N4-Methylcytosine Restriction-Modification System in *Deinococcus radiodurans*

**DOI:** 10.3390/ijms25031660

**Published:** 2024-01-29

**Authors:** Chenxiang Shi, Liangyan Wang, Hong Xu, Ye Zhao, Bing Tian, Yuejin Hua

**Affiliations:** 1MOE Key Laboratory of Biosystems Homeostasis and Protection, Institute of Biophysics, College of Life Sciences, Zhejiang University, Hangzhou 310058, China; 11916015@zju.edu.cn (C.S.); xuhong1685@163.com (H.X.); yezhao@zju.edu.cn (Y.Z.); tianbing@zju.edu.cn (B.T.); 2Cancer Center, Zhejiang University, Hangzhou 310058, China

**Keywords:** *Deinococcus radiodurans*, methyltransferase, restriction endonuclease, R-M system

## Abstract

*Deinococcus radiodurans* is an extremophilic microorganism that possesses a unique DNA damage repair system, conferring a strong resistance to radiation, desiccation, oxidative stress, and chemical damage. Recently, we discovered that *D. radiodurans* possesses an N4-methylation (m4C) methyltransferase called M.DraR1, which recognizes the 5′-CCGCGG-3′ sequence and methylates the second cytosine. Here, we revealed its cognate restriction endonuclease R.DraR1 and recognized that it is the only endonuclease specially for non-4C-methylated 5′-CCGCGG-3′ sequence so far. We designated the particular m4C *R.DraR1*-*M.DraR1* as the DraI R-M system. Bioinformatics searches displayed the rarity of the DraI R-M homologous system. Meanwhile, recombination and transformation efficiency experiments demonstrated the important role of the DraI R-M system in response to oxidative stress. In addition, in vitro activity experiments showed that R.DraR1 could exceptionally cleave DNA substrates with a m5C-methlated 5′-CCGCGG-3′ sequence instead of its routine activity, suggesting that this particular R-M component possesses a broader substrate choice. Furthermore, an imbalance of the DraI R-M system led to cell death through regulating genes involved in the maintenance of cell survival such as genome stability, transporter, and energy production. Thus, our research revealed a novel m4C R-M system that plays key roles in maintaining cell viability and defending foreign DNA in *D. radiodurans*.

## 1. Introduction

Over the course of the evolutionary arms race between bacteriophages (phages) and bacteria, bacteria have developed various defense mechanisms: CRISPR-Cas [1], restriction-modification (R-M) and other related systems [2,3,4,5,6], abortive infection (Abi) mechanisms [7], cyclic oligonucleotide-based anti-phage signaling (CBASS) system [8], and other mechanisms collectively referred to as the pan-immune system of bacteria [9]. The R-M system, comprising DNA methyltransferases (MTases) and restriction endonucleases (REases), constitutes a crucial component of the bacterial immune system and is found in over 74% of prokaryotes [3]. R-M systems are categorized into four types, based on their subunit composition, recognition site, cofactor requirement, and cleavage position [10]. Among them, Type II R-M systems exhibit the widest distribution. Type II REases typically rely on Mg^2+^ for their activity and do not necessitate ATP [11]. The MTases of Type II R-M systems employ S-adenosyl methionine (SAM) as a methyl donor, and both the cognate REase and MTase recognize the same 4–8 base pair DNA sequence. The GIY-YIG family represents a highly conserved type II REase family [12]. The conserved “GIY-YIG” motif is situated in the N-terminal region of these proteins, and specifically recognizes the 5′-CCGC/GG-3′ sequence, including R.Eco29kI [12] and R.Cfr42I [13].

Most DNA MTases act in concert with their corresponding restriction REases to form R-M systems, while a few exist as orphan MTases [14]. MTases are responsible for catalyzing the methylation modification of DNA or RNA in bacteria, and play crucial roles in various epigenetic regulation processes, including chromatin replication and segregation, DNA-protein interactions, DNA mismatch repair, and gene expression regulation [15,16,17]. Bacterial DNA methylation modifications primarily encompass three types: N6-adenine methylation (m6A), C5-cytosine methylation (m5C), and N4-cytosine methylation (m4C) [18]. In the past, the detection of m4C modification was challenging due to limitations in sequencing technologies, leading research to primarily focus on m6A and m5C methylation. However, recent advancements in single-molecule real-time sequencing (SMRT-seq) [19,20] and nanopore-based sequencing [21] have facilitated the identification of bacterial genomic DNA methylation patterns, particularly m4C methylation. 

*Deinococcus radiodurans* is renowned for its remarkable ability to withstand extreme radiation and oxidative stress, which can partly be attributed to its multiple genome copies and distinctive DNA repair mechanisms [22]. In previous studies, we have provided evidence for the existence of a novel α-class N4-cytosine methyltransferase, M.DraR1 [23]. M.DraR1 has been observed to methylate and protect the 5′-CCGCGG-3′ sequence. Deletion of *M.DraR1* leads to differential expression of more than one-third of the organism’s genes and genomic instability. 

In this study, we verified that R.DraR1 can also recognize and cleave the palindromic sequence 5′-CCGC/GG-3′ and form a potential m4C R-M system together with *M.DraR1* within *D. radiodurans*, called DraI R-M system. The DraI R-M system plays a crucial role in regulating the rates of recombination and transformation in this bacterium, and a disruption in the balance of the system can lead to cell death. In vitro experimental results demonstrated distinct protective effects of REases from m4C R-M systems on various methylation substrates, and the cleavage activity of REases specific to m4C R-M systems are hindered by m4C methylation modifications at the 5′-CCGC/GG-3′ sequence, but not m5C methylation modifications. Furthermore, knocking out *M.DraR1* resulted in significantly decreased survival rates, while further knockout of the *R.DraR1* rescued this reduction, and overexpression of *R.DraR1* in vivo resulted in this reduction. There are 44 differentially expressed genes in both of the *ΔR.DraR1* and *ΔM.DraR1* strains. These genes are involved in critical biological processes, such as DNA damage response, transporter, energy production, and biosynthetic pathways, suggesting that the DraI R-M system plays a significant role in maintaining genome stability and essential life processes in *D. radiodurans*. These findings, coupled with the scarcity of m4C R-M systems, suggest the significance of these systems in maintaining genome stability and defending against foreign DNA in *D. radiodurans*.

## 2. Results

### 2.1. A m4C Restriction-Modification System DraI Existed in Deinococcus radiodurans

Recently, we performed a whole-genome sequencing of *D. radiodurans* R1 and revealed a novel α-class N4-cytosine methyltransferase M.DraR1 [23], which can specifically recognize the 6 bp sequence 5′-CCGCGG-3′. Subsequently, we found that plasmids carrying the unmethylated 5′-CCGCGG-3′ sequence exhibited significantly lower transformation efficiency compared to those without this motif. For the majority of Type II R-M systems, the MTases recognize and methylate DNA fragments containing recognition sites, whereas the cognate REases cleave the identical unmethylated sequences [24]. Hence, transformation efficiency assay for DNA fragments containing recognition sites is commonly employed to characterize the biological function of RM system in vivo [25]. When the recognition site is methylated by MTases, the exogenous DNA substrates are not cleaved by the cognate REase, leading to a significantly higher transformation efficiency. We hypothesized that the *D. radiodurans* encodes a defense system involving M.DraR1, such as an R-M system, which could discriminate self and foreign DNA, especially when the DNA donors have a 5′-CCGCGG-3′ site. In scrutinizing the upstream and downstream sequences of *M.DraR1*, we noticed a potential GIY-YIG restriction endonuclease (ORF 15945P) located at the 10th ORF upstream of M.DraR1 (Figure 1A). Its sequence shows high similarity to the *Escherichia coli* restriction endonuclease R.Eco29KI (Figure 1B), which could recognize and cleave the sequence 5′-CCGC/GG-3′ (“/” marks the cleavage position) [13,26]. We named the hypothetical gene *R.DraR1*, while designating the potential m4C R-M system as the DraI R-M system. 

Surprisingly, the m4C methyltransferase *M.DraR1* only exists in the *D. radiodurans* of *Deinococcus* species [27], while homologous genes of the cognate REase *R.DraR1* only exist in *Deinococcus budaensis* (41.36% identity) and *Deinococcus marmoris* (40.72% identity) among other *Deinococcus* strains, which raises our curiosity about the origin of DraI. To further investigate this question, a search for a R-M system that specifically recognizing 5′-CCGCGG-3′ sequence was conducted searching the REBASE database [28]. It has been observed that sequence characteristics enable an easy differentiation between m4C and m5C MTases; although both types of MTases share a conserved N-terminal SAM binding FxGxG motif, m4C MTases harbor the distinct C-terminal catalytic SPPY motif, whereas m5C MTases possess conserved PC, ENV, QRR, IX, and other C-terminal catalytic motifs without SPPY [29]. Totally, 439 REases and 361 MTases were identified, resulting in the identification of 310 potential R-M systems capable of recognizing the 5′-CCGCGG-3′ sequence. However, all of the 310 R-M systems that were identified as m5C R-M systems, their sequence organization varied from each other; most only possess REases and MTases, and some additionally contain V elements (a Vsr-like G/T DNA mismatch repair endonuclease) [30], C elements (an REase gene transcriptional control protein) [31] and H elements (Type II Helicase Domain Protein) (Figure 2A). 

An investigation was conducted using TBtools [32] on the remaining 129 REases and 51 MTases that could not be assigned to specific R-M systems, revealing that the majority of these enzymes are orphans [14], except one putative m4C R-M system in *Roseiflexus castenholzii* (*Rca*). In addition, using R.DraR1 and M.DraR1 as queries, homology searches were performed with the REBASE database Blast tool, resulting in the identification of 590 REases and 2001 MTases. A putative m4C R-M system was annotated in *Roseiflexus species RS-1* (*Rsp*) by this method. Interestingly, this R-M system exhibits a rare pattern where the REase and MTase are fused within the same ORF, which is uncommon among various R-M systems [33,34]. Hence, three putative m4C R-M systems were identified in *D. radiodurans*, *R. castenholzii*, and *R. species* through multiple sequence alignments (see Figure 2B,C). To sum up, there are 310 putative m5C R-M systems distributed among 91 species, indicating that the discovered *M.DraR1-R.DraR1* pair in *D. radiodurans* is a rarity m4C R-M system.

### 2.2. The m4C R-M System DraI Inhibits Foreign Plasmid and DNA Fragment Acquisition

To confirm the function of the m4C R-M system in *D. radiodurans*, the single mutants Δ*R.DraR1* and the double mutant Δ*R* + Δ*M* were constructed (Appendix A) and confirmed using PCR analysis (Appendix A), respectively, and recombination and transformation efficiencies with foreign plasmids were tested. The non-motif shuttle plasmid pRADK, the point-mutated pRADKm with CCGCGG motif, as well as the specifically motif-methylated plasmid M.pRADKm (Appendix A) were transformed into *D. radiodurans*, respectively. The transformation efficiency of all the mutants were significantly increased compared to the wild type, while the double mutant showed the highest efficiency, approximately 5.7-fold (Appendix A); so, we used the relative transformation rate to measure the actual situation of different DNA substrates intruding the separate strains. The relative transformation rate of the strains harboring pRADKm is 14.6 times lower than that of M.pRADKm, using their respective non-motif pRADK strains as a normalization baseline (Figure 3A). However, there is no significant difference in the relative transformation rate between the *ΔR.DraR1* single strain (Figure 3B) and the *ΔR + ΔM* double knockout strain (Figure 3C), indicating the specific restriction role of *R.DraR1* on non-methylation foreign plasmids. However, the non-motif pRADK and the specifically methylated plasmid M.pRADKm showed a similar or higher relative transformation rate, suggesting that the m4C methylated foreign DNA may be recognized by *D. radiodurans* as self-DNA, thereby allowing its transformation into the cells without restriction.

To validate the aforementioned findings, we conducted a recombination efficiency assay using NS and S fragments that carried a chloromycetin resistance gene (See Section 4.4 and Appendix A). The outcomes were in accordance with the results of the transformation efficiency assay above. The DNA fragments exhibited increased susceptibility to degradation by the *D. radiodurans* upon entry into the cells, resulting in a significant difference in absolute recombination efficiency between different strains (Appendix A). Using its own absolute recombination efficiency of NS fragment as the normalization baseline in each mutant strain, the relative recombination rate transformed with the S fragment is significantly 7.6 times lower than that of the ^4m^S fragment in wild-type strains (Figure 3D), but not in the *ΔR.DraR1 or ΔR + ΔM* strains (Figure 3E,F). The results above indicated that DraI R-M system plays an important role in natural transformation processes of the *D. radiodurans*.

### 2.3. R.DraR1 of the DraI System Could Specifically Cleave Unmethylated or m5C-Methylated 5′-CCGC/GG-3′ Sequences In Vitro

The R.DraR1 protein was purified and found to be a homodimer (Appendix A), which is similar to the GIY-YIG family protein, R.Eco29kI [13], but is different from the R.Cfr42I that forms a homotetramer [12]. It was shown that R.DraR1 can cleave DNA substrates containing a 5’-CCGCGG-3’ sequence efficiently (Figure 4A). Any single-base mutation of C/G to T within the CCGCGG sequences of the DNA substrates prevents cleavage by R.DraR1, indicating the specific cleavage activity of R.DraR1 (Figure 4B).

Considering the metal ions in the actual physiological environment, Mg^2+^, Mn^2+^, Ca^2+^, Zn^2+^, and Ni^2+^ were selected to investigate their catalytic effects on the cleavage activity of R.DraR1. It is shown that Mn^2+^ exhibited the most enhancing effect on R.DraR1, whereas Mg^2+^ had minimal catalytic effect on R.DraR1, differing from most REases (Figure 4C). Additionally, unlike other GIY-YIG family proteins [12,35], Ca^2+^, Zn^2+^ and Ni^2+^ were unable to catalyze the cleavage activity (Figure 4D), suggesting a narrower selection range of metal ion cofactors.

The cleavage activity of R.DraR1 can be impeded by methylation introduced by M.DraR1 (Figure 4E,F). However, the m5C methylation-mediated protective effect on the cleavage site is much weaker than that mediated by the m4C methylation modification (Figure 4E,F and Appendix A). These findings suggest that R.DraR1 exhibits greater flexibility in DNA cleavage sites, indicating the broader cleavage ability of REases from the m4C R-M systems, which is beneficial for *D. radiodurans* to better defend against the invasion of foreign DNA.

### 2.4. Imbalance in the DraI System Leads to a Significant Increase in Endogenous Cell Death

Recent research has revealed that REase-induced genomic damage cause DNA damage within an organism, triggering the activation of the SOS system for repair, and programmed cell death can occur in cases of severe damage [36]. *D. radiodurans* does not have the SOS system; nonetheless, we still observed morphological features resembling cell death from the *ΔM.DraR1* strain using Dil and DAPI probes, such as discontinuous extracellular structures, extensive vesicles and flagella, cell swelling, and loss of genetic material (Figure 5A). These particular observations have not been previously reported yet. Cells in the tetraploid form were in a stationary phase and exhibited flagella-like and vesicle-like components, accompanied by cell membrane rupture, indicating that these cells were entering the death phase. The growth rates of different strains were similar from the early to stationary phases, and the colony densities reached a similar level during the stationary phase. However, when the strains were further cultured at 30 °C for 1–7 days after reaching the stationary phase, only the *ΔM.DraR1* strain exhibited a rapid decrease in colony density (Figure 5B). Given that we continuously cultured the cells in a nutrient rich TGY medium without any other environmental stressors, it is highly likely that the observations are the result of endogenous cell death caused by the disruption of certain dynamic balances within the cells.

Then, we utilized the membrane DiBAC4(3) probe and YP1-PI probe to determine the ratio of death cells in different strains. The depolarization of the cells increased while the process of cellular programmed death deepened, as indicated by the stronger signal of the DiBAC4(3) probe (Figure 5C). During the exponential and stationary phases, the membrane potential of the *ΔM.DraR1* strain was always higher than that of other strains, but this enhanced damage disappeared upon *R.DraR1* knockout. The YP1-PI probes rely on detecting membrane permeability and classifying the signal image into necrotic cells (PI-stained), dead (double-stained), normal and apoptotic (YP1-stained) (Appendix A). Cells stained with either YP1 or PI are considered damaged. There were no significant differences during the exponential phase; however, in the stationary phase, the *ΔM.DraR1* strain exhibited a 2.91-fold in the percentage of damaged cells compared to the wild type, while other strains shared no significant difference (Figure 5D). This can be attributed to viable apoptotic cells detected by the FITC channel (Figure 5E), while the *ΔM.DraR1* strain exhibited a 5.64-fold in the percentage of viable apoptosis cells compared to the wild type, suggesting that *D. radiodurans* lacks remedial mechanisms and the programmed cell death process was triggered in the absence of the *M.DraR1*. However, the percentage of viable apoptosis cells were not significantly distinct from the wild type in cells lack of its cognate REase R.DraR1, highlighting the methylation-dependent protection from cleavage of the R-M system.

To further validate the association between endogenous cell death and DraI R-M system imbalance, we achieved controlled over-expression of *R.DraR1* in vivo by employing IPTG-regulated promoters to examine the phenotype. There is no endogenous LacZ activity in *D. radiodurans*. Upon transformation with the IPTG-regulated plasmid pRADKIS-R.DraR1, β-galactosidase activities were significantly increased under 0.5 mM IPTG induction (Figure 5F) (Appendix A). Except for the *ΔM.DraR1* strain in the stationary phase as a positive control, all other strains were cultured to the exponential phase (OD_600_ = 0.6), and the ratio of YP1-PI probe staining, and DiBAC4(3) probe staining before and after IPTG induction separately were measured. The results demonstrated a significant increase in the staining ratio upon IPTG induction. Overexpression of *R.DraR1* caused a 1.19-fold increase in the ratio of positive DiBAC4(3)-stained cells (Figure 5G), and a 15.53-fold increase in the percentage of viable apoptotic cells compared to the wild type (Figure 5H). This confirmed that the imbalance of the novel R-M system triggers cell death events in *D. radiodurans*. However, both increases are slightly (*p* = 0.232 with percentage of viable apoptotic cells, and *p* = 0.135 with percentage of the ratio of positive DiBAC4(3)-stained cells) lower than observed in the *ΔM.DraR1* strain, which could be due to differences in gene expression between introducing the plasmid and the genomic expression. In conclusion, overexpression of *R.DraR1* resulted in a significantly increased proportion of cell death but lower than that of *ΔM.DraR1*, highlighting the significant inhibitory effect of MTase against cleavage by its cognate REase in vivo, emphasizing the importance of balance in the R-M system.

## 3. Discussion

It is well known that the R-M system functions as a barrier against the uptake of foreign DNA [3]. In this study, we identified a novel m4C R-M system, named the DraI R-M system, through multiple sequence alignments, recombination, and transformation efficiency assays. Since the phage of *D. radiodurans* has not been acquired yet, further research is needed to investigate the relative resistance.

A thorough search of the REBASE database was conducted to search the origin of the m4C R-M system that recognizes 5′-CCGCGG-3′ sequences. Only three R-M pairs belonging to m4C R-M systems were predicted so far (REBASE, accessed on 1 December 2023), while the remains are putative m5C R-M systems. No V, C, or H elements are present in the putative m4C R-M systems, implying these three elements might not be necessary for the m4C R-M system (Figure 2B). Further investigation into the m4C R-M system is necessary to determine whether there are specific elements associated with it. The proportion of m4C R-M systems is 0.96% (Figure 2D), highlighting the uniqueness of the m4C R-M system discovered in our study. With the development of sequencing technologies such as single-molecule real-time sequencing (SMRT), an increasing number of m4C modifications will be detected, providing more accurate insights into the genomes of various species [19,20], which is helpful to elucidate the origin of DraI R-M system.

The DraI R-M system exhibits a defensive effect on incoming foreign DNA substrates containing 5′-CCGCGG-3′ sequences. In addition, DNA substrates with methylated sequences exhibited similar or higher relative transformation rates (range from 1.04 to 1.99) compared to non-sequence DNA substrates (Figure 3A–F), suggesting that methylated DNA substrates might be recognized by *D. radiodurans* as self-DNA, emphasizing the importance of *M.DraR1* in the R-M system, as it protects the host genome against cleavage by cognate REases. Furthermore, in the *R1* and *ΔR.DraR1* strains where *M.DraR1* functions normally, we observed a higher relative transformation rate (range from 1.73 to 1.96) for methylated sequence DNA substrates compared to non-sequence DNA substrates (Figure 3A,B,D,E). However, the relative transformation rates for both DNA substrates are similar in the *ΔR + ΔM* strain (range from 1.06 to 1.19) (Figure 3C,F), suggesting that other genes in the DraI m4C R-M system are required to assist in recognizing m4C methylation marks. Nevertheless, further investigation is still required to uncover the specific mechanism.

*R.DraR1* exhibits cleavage activity for broader DNA substrates, including unmethylated and m5C-methylated DNA substrates with a 5′-CCGCGG-3′ sequence, while other homologous REases like R.Eco29kI are hindered by both m4C and m5C modifications mainly due to steric hindrance [37], emphasizing the importance of the m4C DraI system in preserving genome stability and defending against foreign DNA in *D. radiodurans*. Further investigation into the protein structure is needed to determine the precise mechanism.

Unlike other GIY-YIG family proteins, such as R.Cfr42I [12] and UvrC [35], R.DraR1 exhibits its cleavage activity in the presence of either Mn^2+^ (high efficiency) or Mg^2+^ (low efficiency), rather than Ca^2+^, Zn^2+^, and Ni^2+^, suggesting that the structural domain of R.DraR1 is more suitable to Mn^2+^ than other divalent metal ions, which expends the ion preference within the GIY-YIG protein family. The adaptation of R.DraR1 to Mn^2+^ in *D. radiodurans* may be attributed to long-term evolution and a relatively high concentration of Mn^2+^ ions participating in vital cellular activities [38]. Further investigation is needed to gain a more comprehensive understanding of the molecular mechanisms as well as the evolutionary history of m4C R-M systems.

The imbalance of the R-M system results in post-segregational killing in the host cell [36,39,40]. However, due to the thick cell wall of *D. radiodurans* [41], the inner membrane may not easily detect when the cell undergoes death. We observed a significant increase in cell death of the *ΔM.DraR1* strain, where only *R.DraR1* was expressed, while heightened damage was diminished upon the knockout of *R.DraR1*. Overexpression of *R.DraR1* confirmed its role in the significant post-segregational killing events in vivo. It was previously suggested that the imbalance of the R-M system triggers programmed cell death through the apoptosis pathway [36], relying on the involvement of the SOS pathway [42], which is absent in *D. radiodurans*. Although the specific mechanism is still under investigation, similar programmed cell death has been observed in the *ΔM.DraR1* strain, suggesting that the imbalance of DraI system induces endogenous cell death through potential stress-response pathways within cells.

We identified 104 differentiated expressed genes (DEGs) by comparing the transcriptome of the *ΔR.DraR1* vs. *R1* and found that the expression levels of the 44 DEGs were also significantly changed in *ΔM.DraR1* (Figure 6). The majority of these DEGs (40 out of 44) exhibited opposite directions of upregulation or downregulation in the two mutant strains, implying putative interactions between the DraI R-M pair in vivo (Table 1). Both *ΔR.DraR1* and *ΔM.DraR1* exhibited DEGs of the DNA damage response genes [43], suggesting that an imbalance in the DraI R-M system can induce DNA damage similar to external stresses, thereby significantly impacting the genome stability of cells. Particularly, *D. radiodurans* possesses a unique DNA damage repair system composed of PprI and Ddr protein families, indicating an unusual DraI R-M regulation mechanism in this bacterium. Ion transport and metabolism, energy synthesis and transfer, and several other pathways are also greatly affected by the *R.DraR1* or *M.DraR1* deletion, suggesting that an imbalance in a DraI R-M system could lead to a comprehensive change in the vital biological processes. However, further investigation is needed to understand the underlying mechanism.

## 4. Materials and Methods

### 4.1. Bacterial Strains and Culture Conditions

All *D. radiodurans* R1 strains used in this study are summarized in Appendix A. All of the *D. radiodurans* strains were grown in tryptone glucose yeast extract (TGY) liquid media, or on agar plates (TGY liquid media with additional 1.5% Bacto-agar). *E. coli* strains were grown in Luria-Bertani (LB) liquid media or on agar plates, supplemented with the appropriate antibiotics at 37 °C with aeration.

### 4.2. DNA Manipulation

We used tripartite ligation and double-crossover recombination methods for gene mutation [44]. Briefly, about 1000 bp upstream and downstream fragments of the target gene were ligated to a streptomycin, kanamycin, or chloromycetin resistance cassette containing homologous arms by overlapping PCR. Mutant colonies were screened on TGY plates containing 40 µg/mL kanamycin, 20 µg/mL streptomycin, and 5 µg/mL chloromycetin, then confirmed by PCR and DNA sequencing. The mutant strains were named *ΔR.DraR1*, *ΔR + ΔM*. All primers are listed in Appendix A.

### 4.3. Database Search for R-M Systems Which Recognize 5′-CCGCGG-3′ Sequence

All sequence data were obtained from the REBASE database (http://rebase.neb.com/rebase/index.html, accessed on 1 December 2023), while genomic sequences were downloaded from NCBI. TBTools was utilized to perform whole-genome Blast searches and partial sequence alignments [32]. Briefly, the process consists of searching for homologous protein genes of R.DraR1 or M.DraR1 using REBASE, followed by their mapping to the genome. Subsequently, the presence of the corresponding REase or MTase in the gene’s genomic location is verified. If present, a sequence alignment is conducted with R.DraR1 or M.DraR1. Ultimately, the relative length of the newly predicted R-M system should be within 10 ORFs.

### 4.4. Transformation and Recombination Efficiency Assays

The *E. coli*-*D. radiodurans* shuttle vector pRADK was used to perform transformation efficiency assays, which confers ampicillin resistance in *E. coli* and chloromycetin resistance in *D. radiodurans*. A total of 0.2 µg of each of the shuttle vector pRADK, unmethylated and methylated pRADKm [23] was transferred into *D. radiodurans* strains. The transformed cells were plated onto TGY and TGY containing chloromycetin (TC) plates, and the transformation frequency was determined by dividing the number of transformant colonies growing on TC plates by the total number of viable cells on the TGY plates [45]. A total of 0.5 µg of the donor DNA fragment containing a chloromycetin resistance gene from *ΔcrtB* mutant [46] (Named NS fragment, no CCGCGG site inside resistance gene sequence) and its counterpart with an additional unmethylated or methylated CCGCGG site just after the start codon (Named S fragment, one CCGCGG site) were used to measure the recombination capacity of the mutants. All fragments were inserted into pUC19 (Appendix A). Four replicates were performed for each strain.

### 4.5. Expression and Purification of R.DraR1

The full-length R.DraR1 gene was amplified and cloned into a modified pET28a expression vector, which placed the coding sequence behind an N-terminal SUMO tag and Ulp1-cleavable site. After its confirmed by DNA sequencing, the constructed plasmid was transformed into an *E. coli* ER2566 strain. An expression of the R.DraR1 protein was induced overnight or at least 20 h at 16 °C, by adding isopropyl-β-D-thiogalactopyranoside (IPTG) to a final concentration of 0.4 mM.

The cells were harvested by centrifugation and resuspended in a lysis buffer (20 mM Tris-HCl, pH 8.0; 500 mM NaCl; 5% (*w*/*v*) glycerol), then lysed by sonication, and clarified by centrifugation at 14,500 rpm. The supernatant was purified using a HisTrap HP column (GE Healthcare, Chicago, IL, USA) equilibrated with buffer A (20 mM Tris-HCl pH 8.0; 500 mM NaCl; 5% (*w*/*v*) glycerol), washed with 50 mM imidazole and eluted with 250 mM imidazole. After the SUMO tag removal using Ulp1 protease, the composition was desalted using a HiPrepTM 26/10 column (GE Healthcare) with buffer A to rid of the imidazole and was purified using a HisTrap HP column (GE Healthcare) equilibrated with buffer A again, but the flow-through fractions were collected. Finally, the protein was purified using a Superdex 200 10/300 GL column (GE Healthcare) with buffer B (20 mM Tris-HCl, pH 8.0 and 200 mM KCl), and assessed by 12% sodium dodecyl sulfate (SDS)-polyacrylamide gel electrophoresis (PAGE).

### 4.6. In Vitro M.DraR1 Protection and R.DraR1 Cleavage Assays

The M4C MTase M.DraR1 assays were carried out at 30 °C with a total volume of 50 μL of reaction mix, while m5C MTase M.SssI (NEB Biotech, Ipswich, MA, USA) assays were carried out at 37 °C. Substrate DNA were incubated with MTase for 2 h in reaction buffer 1 (20 mM Tris-HCl, pH 8.0; 100 mM KCl; 1 mM DTT; 3 mM β-mercaptoethanol; and 160 µM S-adenosyl-l- methionine, SAM) or NEB buffer 2.

Endonuclease R.DraR1 cleavage assays were carried out at 30 °C with a total volume of 10 μL of reaction mix. Substrate DNAs were incubated with REase for 2 h in reaction buffer 2 (20 mM Tris-HCl, pH 8.0; 100 mM KCl; 1 mM DTT) with 10 mM manganese chloride. Reactions were terminated by the addition of a stop solution (50 mM EDTA, pH 8.0, 50% glycerol, 0.02% bromophenol blue) and heated to 65 °C for 15 min. Reaction products were analyzed using 1% agarose gels or 20% UREA-PAGE gels.

Cleavage reactions with Mg^2+^, Mn^2+^ or Ca^2+^ metal-ion cofactors were initiated by addition of divalent metal ions (MgCl_2_, MnCl_2_ and CaCl_2_) to a final concentration of 1 or 10 mM, and extra NaCl were supplemented to equalize the ionic strength. Except for the metal-ion cofactors, other components in reaction buffer 2 were not changed. Reactions were also terminated and analyzed with 20% UREA-PAGE gels.

### 4.7. Fluorescence Microscopy and FACS Analysis

The nuclear stain DAPI (Beyotime Biotech, Shanghai, China) and the membrane stain Dil (Beyotime Biotech) were utilized to label cells of different time points of *D. radiodurans* strains in order to observe and analyze cell morphology. DAPI (blue) can permeate the cell wall and label the genomic DNA, with a maximum excitation wavelength of 364 nm and a maximum emission wavelength of 454 nm. Dil (red) binds to components with a phospholipid bilayer membrane and labels them with a maximum excitation wavelength of 549 nm and a maximum emission wavelength of 565 nm. Each strain used for staining was cultured until it reached the exponential phase (OD_600_ = 1.0) or the stationary phase (30 °C for 24 h). Subsequently, 1 mL of the bacterial culture was centrifuged at 5000 rpm for 5 min to collect the bacterial cells. After washing with PBS, 0.1 mL of PBS was added to resuspend the cells, followed by the addition of 10 μL of the stain. Dil staining was conducted for 20 min, after which DAPI was added for an additional 5 min of staining. Following the staining process, 3 μL of the stained cells were placed on a glass slide for observation using an inverted fluorescence microscope Elipse Ti2 (Nikon, Tokyo, Japan) under a 100× objective.

The YP1-PI Kit (Beyotime Biotech), which is based on permeability detection, and the DiBAC4(3) probe (UElandy Biotech, Suzhou, China), used for detecting membrane potential, were employed for fluorescence microscopy confirmation and FACS analysis. Each strain was cultured until it reached the appropriate stage, and 0.5–1 mL of the bacterial culture was collected through centrifugation. After washing with PBS, the cells were incubated with the respective probe at a final concentration of 1× at room temperature for 25 min. The staining percentage was confirmed using a fluorescence microscope, and once confirmed, the samples were diluted to a final volume 1000 μL. Fluorescence detection was conducted using the CytoFlex flow cytometer (Beckman Coutler life sciences, Indianapolis, IN, USA). The negative controls with unstained cells were utilized, and compensation was adjusted for single-stained cells. The proportions of the cells in each region were determined and calculated (n = 3; 200,000 cells were detected for each experience), and the FlowJo software v 10.6.2 was used for compensation.

### 4.8. Over-Expression of R.DraR1 and β-Galactosidase Assay

*D. radiodurans* does not possess an IPTG-LacI expression system. We implemented the method outlined by François L. et al. [47] to introduce LacI at the AmyE locus in the genome. Additionally, we modified the shuttle plasmid pRADK to serve as an expression vector for R.DraR1 in vivo. First, we amplified the fusion fragment of the strong promoter pGroES, LacI, and the tetracycline resistance gene (tetR) used for screening through overlap-PCR. Then, we employed the tripartite ligation and double-crossover recombination techniques to replace the AmyE region with the fragment. Subsequently, LacI was inserted at the HindIII and XhoI site of the pRADK plasmid. Furthermore, the original pGroES promoter in the MCS region of pRADK was replaced with the LacI-repressed pSpac promoter derived from pMUTIN4. Additionally, we inserted LacZ into the MCS region of the pRADKIS plasmid to evaluate the system’s functionality using the β-galactosidase assay [48]. Finally, we employed the pRADKIS-R.DraR1 plasmid for over-expression in *D. radiodurans* in vivo.

### 4.9. RNA Isolation and Sequencing Analysis (RNA seq)

A RNA sequencing analysis was performed, as in [23]. Briefly, total RNA was isolated using the Trizol method. The 1% denaturing agarose gels were used to assess RNA degradation and contamination. RNA concentration and RNA integrity were measured using an RNA Nano 6000 Assay kit and Bioanalyzer 2100 system (Agilent Technologies, Santa Clara, CA, USA). A total of 3 G RNA text per sample were used as the input material for library preparation. The library preparations were sequenced on an Illumina Hiseq platform and paired-end reads were generated. The raw reads were deposited in the Sequence Read Archive (SRA) database of NCBI (PRJNA1054005).

### 4.10. Statistical Analysis

To minimize randomness and determine the range of the experimental data, we conducted preliminary experiments after consulting similar studies. Then, we examined the data distribution and subsequently performed at least 3 duplicates based on the experimental conditions and the range of counts obtained from the preliminary experiments. From these duplicates, we selected 3 to 4 sets of data that approximately conformed to the gaussian distribution instead of being discrete. Data were collected and then analyzed using GraphPad Prism software v 7.0. Data were represented as ±SD values of pooled experiments. *p*-Values were calculated using *t*-Test, one-way ANOVA or two-way ANOVA. N.S.: *p* > 0.05, * *p* < 0.05, ** *p* < 0.01, *** *p* < 0.001.

## Figures and Tables

**Figure 1 ijms-25-01660-f001:**
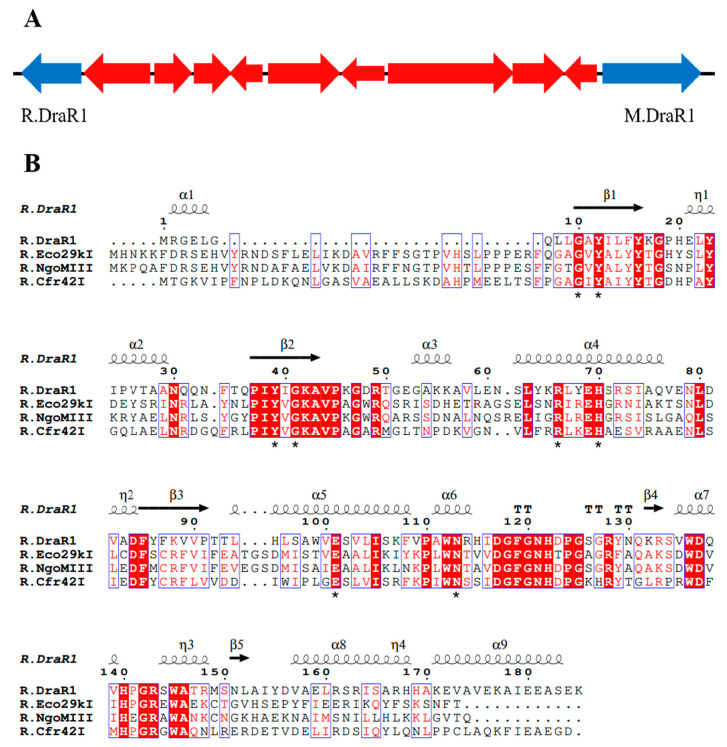
The locus of a m4C restriction-modification system and alignments of *R.DraR1* in *D. radiodurans*. (**A**) Locus encoding the predicted R-M system. (**B**) Multiple sequence alignments of *R.DraR1* and other reported Type-IIE GIY-YIG endonucleases using ClustalW (https://www.genome.jp/tools-bin/clustalw, accessed on 5 October 2023) and ESPript (https://espript.ibcp.fr/ESPript/cgi-bin/ESPript.cgi, accessed on 5 October 2023). The conserved GIY-YIG core and residues that are critical for its catalytic and formation of active site are indicated by asterisks below (*).

**Figure 2 ijms-25-01660-f002:**
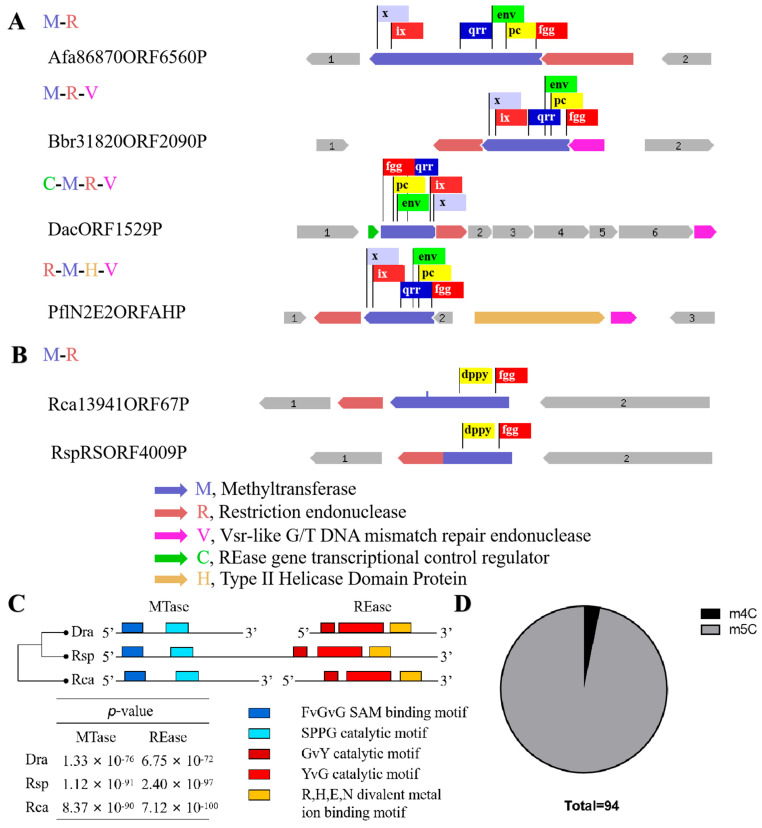
Different sequence organization of R-M system which recognize CCGCGG sequence. (**A**,**B**) Boxes of various colors represent various conservative domains. (**A**) Some representative sequence organization of putative m5C R-M systems. (**B**) Organization of putative m4C R-M system apart from DraI. (**C**) Multiple sequence alignments of three m4C R-M systems. Motif prediction was using MEME (https://meme-suite.org/meme/, accessed on 1 December 2023), distance clustering plot of 3 putative R-M system was using ClustalW. The five different colored blocks represent different conserved domains. (**D**) Proportion of m4C (3/94) and m5C R-M system.

**Figure 3 ijms-25-01660-f003:**
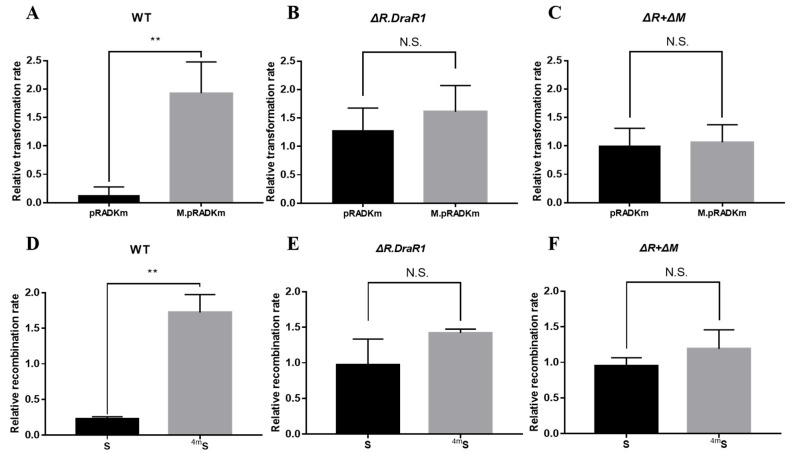
A m4C restriction-modification system inhibits plasmids and fragments acquisition. (**A**–**C**) The relative transformation rate of *R1*, *ΔR.DraR1* and double mutant *ΔR + ΔM*. The transformant counts when transformed with a non-motif vector pRADK, unmethylated vector pRADKm with a CCGCGG motif or a methylated vector M.pRADKm with a CCGCGG motif. As well, 0.2 µg plasmids were transformed into each strain. (**D**–**F**) The relative recombination rate of *R1*, *ΔR.DraR1* and *ΔR + ΔM*. 0.5 µg of the chloromycetin resistance gene fragment carrying one non-methyl site (S) or one methyl site (^4m^S) or not (NS) was transformed as donor DNA into each strain. All of the efficiencies were calculated on TGY plates containing 5 mg/mL chloromycetin, divided by the total number of viable cells, and relative rates were calculated using absolute efficiency of non-motif vector pRADK or NS fragment as the normalization baseline. Data were represented as ±SD values of pooled experiments. N.S. = no significant differences. *p*-Values were calculated using *t*-Test. N.S.: *p* > 0.05, ** *p* < 0.01.

**Figure 4 ijms-25-01660-f004:**
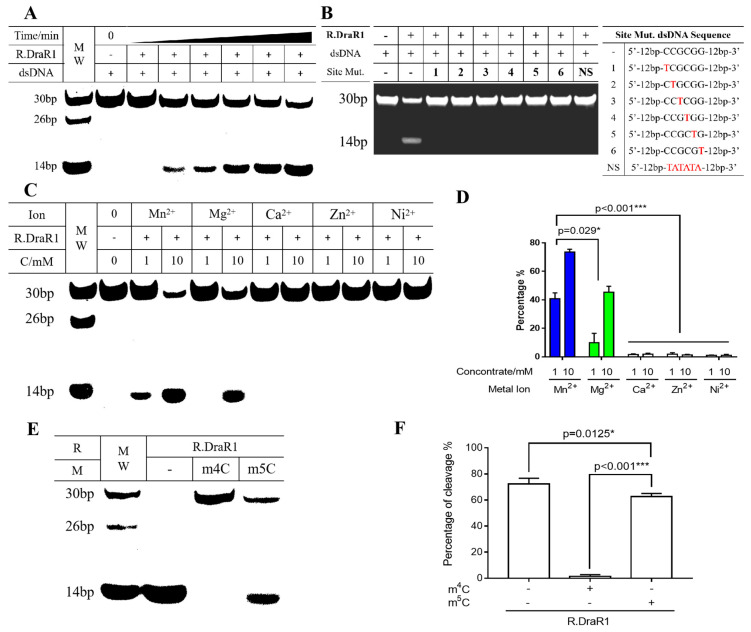
The cleavage of R.DraR1 can be prevented by the specific methylation and protection of the 5’-CCGCGG-3’ sequence by m4C MTase M.DraR1. (**A**,**B**) R.DraR1 could only cleave the DNA substrates with 5’-CCGCGG-3’ sequence. (**A**) Different time of cleavage (ranging from 0 to 90 min) and 250 nM DNA substrate was used. (**B**) 1 mM DNA substrate was added in cleavage buffer (KCl 200 mM, DTT 1 mM, Mn^2+^ 10 mM) at 30 °C for 2 h. The site mutations of DNA substrate from C1-C6 were indicated with red, the whole sequences were shown in Appendix A. (**C**,**D**) The cleavage activity of R.DraR1 is more dependent on Mn^2+^ ions compared with other divalent metal ions. Extra NaCl were supplemented to equalize the ionic strength, resulting in a final ionic strength of 30 mM in each reaction. (**E**,**F**) The REase from the m4C R-M system has a broader range of DNA substrates for cleavage. (**C**–**F**) 100 mM DNA substrate was used in cleavage buffer (KCl 200 mM, DTT 1 mM, Mn^2+^ 10 mM) at 30 °C for 1 h. Data were represented as ±SD values of pooled experiments. *p*-Values were calculated using one-way ANOVA. * *p* < 0.05, *** *p* < 0.001.

**Figure 5 ijms-25-01660-f005:**
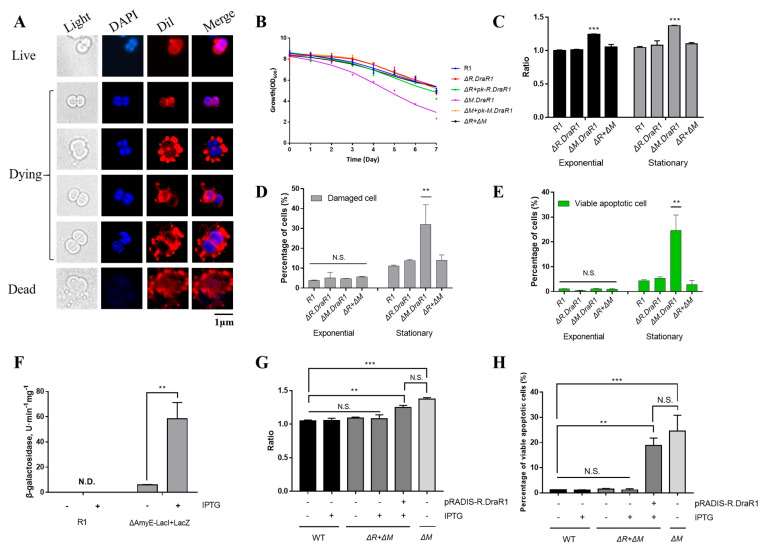
An imbalance in the m4C R-M system leads to a significant increase in cell death. (**A**) Cell morphology of *ΔM.DraR1* mutant. Cells stained with Dil were shown in red color and cells stained with DAPI are shown in blue color. (**B**) Growth curves of *D. radiodurans* strains. The wild-type, mutant, and complementary strains were individually grown in TGY medium, and growth rates were recorded by measuring the OD_600_ every 24 h. (**C**) Ratio of DiBAC4(3) stained cells of *D. radiodurans* strains (n = 3). Cells with DiBAC4(3) intensity > 5000 were considered positive. All of the data were normalized, and the staining ratio of R1 in the exponential phase was used as the baseline. (**D**,**E**) Percentage of stained cells of *D. radiodurans* strains (n = 3). (**F**) β-galactosidase assay was used to verify the functionality of the IPTG-LacI system. (**G**) Percentage of YP1-stained viable apoptotic cells before and after IPTG induction (n = 3). (**H**) Ratio of DiBAC4(3) stained cells before and after IPTG induction (n = 3). Data were represented as ±SD values of pooled experiments. N.S. = no significant differences. *p*-Values were calculated using two-way ANOVA. N.S.: *p* > 0.05, ** *p* < 0.01, *** *p* < 0.001.

**Figure 6 ijms-25-01660-f006:**
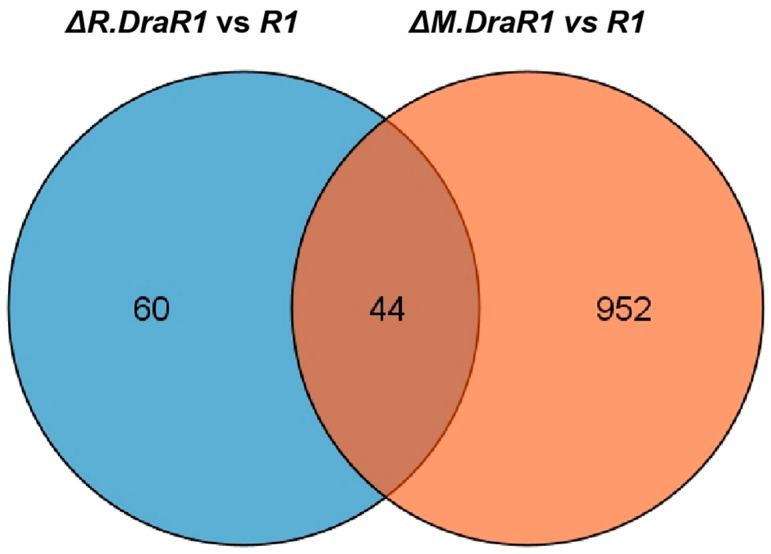
Venn diagram showing the number of unique and shared of DEGs genes in *ΔR.DraR1* vs. *R1* (Blue) compared to *ΔM.DraR1* vs. *R1* (Orange) as determined by RNA-seq.

**Table 1 ijms-25-01660-t001:** Expression changes in genes involved in different pathways in the deficient strains.

Gene ID	Function Annotation	*ΔR.DraR1* vs. *R1*.DEG		*ΔM.DraR1* vs. *R1*.DEG	
		log2FC	*p*-Value	log2FC	*p*-Value
**DNA damage response protein**					
*DR_0423*	Single-stranded DNA-binding protein, ddrA	−1.86	6.56 × 10^−6^	6.61	1.59 × 10^−56^
*DR_0070*	Single-stranded DNA-binding protein, DdrB	−1.47	1.56 × 10^−4^	4.66	5.89 × 10^−27^
*DR_0003*	DNA damage response protein C, ddrC	−2.39	3.07 × 10^−9^	4.36	9.44 × 10^−19^
*DR_0326*	DNA damage response protein D, ddrD	−1.39	1.15 × 10^−3^	5.89	2.11 × 10^−26^
*DR_B0100*	RNA ligase domain-containing protein, DdrP	−1.08	2.34 × 10^−3^	3.14	3.73 × 10^−14^
*DR_2574*	Transcriptional repressor of the RDR regulon, DdrO	−1.28	4.14 × 10^−4^	3.57	3.47 × 10^−27^
*DR_A0346*	DNA repair protein PprA	−1.52	1.53 × 10^−3^	3.13	6.37 × 10^−12^
*DR_C0012*	GerE family transcriptional regulator	2.37	1.40 × 10^−8^	9.35	2.05 × 10^−7^
**Transporter**					
*DR_0203*	ABC transporter permease	−1.18	1.36 × 10^−2^	1.79	6.25 × 10^−5^
*DR_1036*	branched-chain amino acid ABC transporter permease	1.05	2.38 × 10^−3^	−0.96	1.61 × 10^−2^
*DR_1142*	protein transporter	−2.18	1.91 × 10^−10^	1.22	8.28 × 10^−3^
*DR_2612*	undecaprenyl phosphate transporter, DedA	−1.31	1.09 × 10^−2^	1.57	1.18 × 10^−4^
*DR_B0083*	Potassium-transporting ATPase subunit B, kdpB	−1.11	1.82 × 10^−2^	2.58	1.36 × 10^−3^
*DR_B0086*	Potassium-transporting ATPase subunit A, kdpA	−1.32	1.60 × 10^−2^	2.23	4.56 × 10^−4^
*DR_A0073*	cation-transporting P-type ATPase	−1.18	4.98 × 10^−4^	2.05	1.12 × 10^−9^
**Energy production**					
*DR_A0076*	AAA+ ATPase domain-containing protein	−1.72	4.25 × 10^−3^	3.48	4.34 × 10^−5^
*DR_C0016*	Phosphodiester glycosidase domain-containing protein	−5.12	7.99 × 10^−25^	4.44	2.27 × 10^−6^
*DR_C0009*	KAP NTPase domain-containing protein	2.08	1.53 × 10^−6^	1.23	1.04 × 10^−2^
*DR_C0023*	N-terminal DNA-binding domain-containing protein, KfrA	−1.27	2.53 × 10^−2^	7.66	3.71 × 10^−4^
**Biosynthetic pathway**					
*DR_0133*	acetyl xylan esterase	−1.00	2.84 × 10^−2^	2.00	5.51 × 10^−7^
*DR_0422*	Trans-aconitate 2-methyltransferase, tam	−1.26	7.85 × 10^−5^	5.17	3.86 × 10^−31^
*DR_0598*	Peptidase M23 domain-containing protein, LytH	−1.05	5.65 × 10^−3^	3.26	5.56 × 10^−3^
*DR_0659*	frnE protein	−1.35	2.57 × 10^−5^	3.15	2.59 × 10^−4^
*DR_A0029*	4-aminobutyrate aminotransferase, argD	−1.20	1.23 × 10^−3^	−1.66	1.33 × 10^−3^
*DR_A0178*	xanthine dehydrogenase C-terminal subunit	1.10	7.00 × 10^−3^	−1.17	1.03 × 10^−3^
*DR_A0179*	xanthine dehydrogenase accessory protein XdhC	1.39	4.09 × 10^−5^	−1.15	4.02 × 10^−3^
*DR_A0180*	guanine deaminase, guaD	1.14	3.86 × 10^−4^	−1.12	3.02 × 10^−3^
*DR_B0060*	Spore coat protein U domain-containing protein	−2.40	9.31 × 10^−6^	6.46	8.75 × 10^−6^
*DR_C0013*	N-acetylmuramoyl-L-alanine amidase	−4.18	4.30 × 10^−14^	6.07	2.26 × 10^−5^
**Nucleoid-associated gene**					
*DR_0906*	DNA gyrase subunit B, GyrB	−1.07	1.61 × 10^−2^	0.89	5.65 × 10^−3^
**Exonuclease**					
*DR_C0006*	putative 3′-5′ ssDNA/RNA exonuclease, TatD	1.56	1.69 × 10^−5^	1.42	5.19 × 10^−3^
**Natural competence regulated gene**					
*DR_2338*	cinA protein	−1.25	1.59 × 10^−3^	2.56	2.18 × 10^−15^
**Peptidase activity**					
*DR_A0028*	DUF4274 domain-containing protein	−1.36	5.85 × 10^−3^	−1.77	4.53 × 10^−4^
*DR_A0283*	S8 serine protease	1.23	1.39 × 10^−2^	−0.96	1.06 × 10^−2^

## Data Availability

The RNA-seq data from this study have been submitted to the NCBI Gene Expression Omnibus with an accession number of GSE251690.

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
