# Peer review of "Characterization of a Novel N4-Methylcytosine Restriction-Modification System in Deinococcus radiodurans"

_ijms, 2024, doi:10.3390/ijms25031660_

Round 1
Reviewer 1 Report
Comments and Suggestions for Authors
This work is well performed experiments and wrote the manuscript. Interestingly, the knocking out M.DraR1 strain resulted in significantly decreased survival rate, while further knockout of R.DraR1 rescued this reduction. Because of 44 differentially expressed genes in both of the ΔR.DraR1 and ΔM.DraR1 strains, these genes are involved in critical biological processes such as DNA damage response, transporter, energy production, and biosynthetic pathways using RNA-Seq analysis and in visual observations. It is really new conceptional finding in a radiation-resistant bacterium. Thus, this MS is well suitable to publishing in IJMS after minor revision.
L64: In this study, We verified > change to In this study, we verified
L86-88: Subsequently, we found that plasmids carrying the unmethylated 5’-CCGCGG-3’ sequence exhibited significantly lower transformation efficiency compared to those without this motif. – transformation frequency depends on a target seq’s methylation? Need to explain the evidences for transformation frequency
L111: enable a easy differentiation between m4C and m5C > change to enable an easy differentiation between m4C and m5C
L144-145: (Supplementary Figure 1 A-D) > Supplementary Fig 1B &D
L163: that carried a chloramphenicol resistance gene / L180: chloromycetin resistance gene – not matched both sentences
L377, 379: chloromycetin? Or chloramphenicol?
L392: pRADK > need to info for shuttle vector in text and Sup. Table 1, especially antibiotic markers.
L394: chloromycetin (TC) plates – where chloromycetin resistance is come from?
- Need to check for antibiotic resistance from this unclear situation.
L185: using t-Test > change to using t-Test
L232-233: the stable phase > change to stationary phase
L238: Then we utilized the the membrane DiBAC4(3) probe > change to Then we utilized the membrane DiBAC4(3) probe
L266: n IPTG induction.Overexpression of R.DraR1 > change to n IPTG induction. Overexpression of R.DraR1
L320: DraI m4C R-M system are requiredto assist in recognizing > change to DraI m4C R-M system are required to assist in recognizing
L364: vital biological processes.However, further > change to vital biological processes. However, further
L373, 408: ◦C > change to ˚C
L407: ER2566 strain > change to E. coli ER2566 strain
L510: Please, revised references following journal guidelines. For examples, journal names, paper titles, and italic forms.
Fig. 2AB – need to better quality control for reading, specially the flags
Fig. 4D – concontrate/ mM change to concentrate/ mM
Fig. 5A: The dying cells were much larger than live cells. Maybe need to explain why cells revealed different size. Dapi > change to DAPI
In supplementary data legends: need to check and revise mistyping.
Sup. Fig. 3. Please added the positions of standard marker proteins for gel filtration data.
Sup. Fig. 4B. Need more clear illustration for construction scheme
Comments on the Quality of English LanguagePlease check carefully and revised misstypo and gramma in text.
Followed journal guidelines for references.
Reviewer 2 Report
Comments and Suggestions for Authors
The manuscript entitled: “Characterization of a novel N4 -methylcytosine restriction-modification system in Deinococcus radiodurans” is a well written manuscript. It is easy to follow, and it comprises a considerable amount of work and describes important insights. These insights are, in my point of view, relevant.
Please allow me to suggest some improvements:
In my opinion the manuscript impact would be considerably improved if the description comprised within line to 35 would be layout in a scheme.
All graphics in Figure 3 should have the same scale in the Y axis.
Figure 5 A does not have a scale. Please revise.
I failed to find the description of the procedures to achieve the Purification and identification of R.DraR1 enzyme. Can the authors please inform where this information is present or please include it.
Materials and Methods should include a subsection comprising the description and details of the statistical analysis performed.
Supplementary Figure 3, the authors state “The purity of R.DraR1 protein is calculated to be >90% using ImageJ software”, can the authors please provide additional details?
Please separate the numerical values from the units, except percentage. Please revise throughout the entire manuscript.
Round 2
Reviewer 2 Report
Comments and Suggestions for Authors
I acknowledge the improvements performed by the authors, however, there are two issues that I would like to clarify with the authors:
1. The authors added: “Data were collected and analyzed using GraphPad Prism software. The data were presented as ±SD values of pooled experiments. P-Values were calculated using t-Test, one-way ANOVA or two-way ANOVA. N.S.: P>0.05, *P<0.05, **P<0.01, ***P<0.001.”, since the authors performed parametric statistical analysis, a description of the tests performed to assess the prerequisites of the data should be included. Namely, the tests to confirm the gaussian distribution of the data.
2. I have doubts what the authors used as control for the, or standard protein concentration during the analysis described in Response 6: “ImageJ software is commonly employed for grayscale analysis to quantify protein purity. To quantify the gel image, it is imported into ImageJ, and the "Analyze > Gels > Select First Lane" command is utilized to designate the lane of interest. Subsequently, the "Analyze > Gels > Plot Lane" command is employed to conduct grayscale quantification. The orginal gel image is below.”
